# Fluorescent PSC-Derived Cardiomyocyte Reporter Lines: Generation Approaches and Their Applications in Cardiovascular Medicine

**DOI:** 10.3390/biology9110402

**Published:** 2020-11-16

**Authors:** Naeramit Sontayananon, Charles Redwood, Benjamin Davies, Katja Gehmlich

**Affiliations:** 1Division of Cardiovascular Medicine, Radcliffe Department of Medicine and British Heart Foundation Centre of Research Excellence, University of Oxford, Oxford OX3 9DU, UK; naeramit.sontayananon@rdm.ox.ac.uk (N.S.); credwood@well.ox.ac.uk (C.R.); 2Wellcome Centre for Human Genetics, Roosevelt Drive, Oxford OX3 7BN, UK; 3Institute of Cardiovascular Sciences, University of Birmingham, Birmingham B15 2TT, UK

**Keywords:** fluorescent PSC-CM reporter lines, transgenic methods, CM purification, cardiac development, optical electrophysiology

## Abstract

**Simple Summary:**

Understanding how heart muscle cells function and developing new strategies for repairing the damaged heart are important challenges for tackling heart disease. Human cells from the heart are required in the laboratory for these investigations, but are difficult to obtain in large numbers from patients. As an alternative, stem cells can be used which can be cultured indefinitely in the laboratory and then turned into heart muscle cells. The current methods lead to a mixture of cell types, belonging to the different regions of the heart, creating difficulties when trying to understand the biology of a particular area of the heart, or for future applications when these stem cell-derived cell types are used to repair the heart. Genetic modification of stem cells provides a solution to this problem, as these techniques allow fluorescent markers, so-called reporters, to be inserted into key genes that are active in the different cell types. The different coloured reporters thus allow the identification and purification of specific cell types. In this review, we discuss the various methods that can be used to establish these reporter systems and highlight their applications in different aspects of cardiovascular medicine.

**Abstract:**

Recent advances have made pluripotent stem cell (PSC)-derived cardiomyocytes an attractive option to model both normal and diseased cardiac function at the single-cell level. However, in vitro differentiation yields heterogeneous populations of cardiomyocytes and other cell types, potentially confounding phenotypic analyses. Fluorescent PSC-derived cardiomyocyte reporter systems allow specific cell lineages to be labelled, facilitating cell isolation for downstream applications including drug testing, disease modelling and cardiac regeneration. In this review, the different genetic strategies used to generate such reporter lines are presented with an emphasis on their relative technical advantages and disadvantages. Next, we explore how the fluorescent reporter lines have provided insights into cardiac development and cardiomyocyte physiology. Finally, we discuss how exciting new approaches using PSC-derived cardiomyocyte reporter lines are contributing to progress in cardiac cell therapy with respect to both graft adaptation and clinical safety.

## 1. Introduction

Pluripotent stem cells (PSCs), including embryonic stem cells (ESCs) and induced pluripotent stem cells (iPSCs), have revolutionised the study of developmental biology and diseases [1,2,3,4,5]. Although animal models provide valuable in vivo insights into developmental processes, they may not recapitulate human biology fully due to species-specific characteristics [6,7,8,9]. PSCs, on the other hand, are obtainable from humans. Human ESCs (hESCs) are derived from preimplantation human embryos, generally donated as a by-product of fertility treatment, which has raised ethical concerns [10]. Human iPSCs (hiPSCs), however, avoid these issues, as they can be reprogrammed from somatic cells [11]. Potential challenges in the use of iPSCs primarily lie in their preparation. The reprogramming process leaves epigenetic relics, e.g., DNA methylation, which may cause variation in gene expression profiles among different iPSC lines and between iPSC and ESC lines [12]. Compared to ESCs, some iPSC lines exhibit differentiation delay or defects [13], partly due to incomplete reprogramming. The choice of iPSC lines needs to be validated for the desired lineages as there may be biases in differentiation capacity due to the somatic origin from which the lines were reprogrammed [14]. In addition, substantial iPSC passaging increases the risk of genome instability and possibly leads to phenotypic variation in iPSC-derived progenies [15]. 

Although primary adult cardiomyocytes (CMs) have provided important insights [16,17], their lack of proliferation and short culture life prevents long-term studies. By contrast, PSCs are pluripotent and can self-renew, making them powerful research tools. They can be cultured indefinitely and can be differentiated into various types of cells, including cardiac lineages. In vitro PSC-CM differentiation involves sequential steps, primarily regulated by the WNT signalling pathway [18]. WNT activation induces PSC transition to mesodermal stages and subsequent inhibition of the pathway steers the fate of the cells towards cardiac mesoderm and cardiac progenitor cells (CPCs), giving rise to both CM and non-CM progenies. PSC-CMs may also have a potential therapeutic application in cardiac tissue repair. However, two limitations complicate such applications. Firstly, PSC-CMs display embryonic characteristics, both structurally and functionally, and secondly, the differentiation protocols yield mixed populations of different CM subtypes, including atrial, ventricular and nodal cells, and non-CM cell types. 

The ability to label individual cell types non-invasively using a genetically encoded fluorescent reporter offers the prospect of greater insight into cardiac differentiation and the isolation of pure populations of specific cell types. In this review, we will discuss how such fluorescent reporter lines are generated and how reporter cells can be used to study developmental cardiology, CM function and how they have advanced cardiovascular medicine. 

## 2. Establishment of Transgenic Fluorescent Reporter Lines

In order to produce a fluorescent reporter line, a transgenic construct is required comprising a fluorescent protein (FP) inserted downstream of a promoter or gene of interest (GOI). A reporter line can be generated as a transient line, where the transgene remains episomal, or as a stable line, where the transgene becomes integrated into the host genome, either site-specifically or randomly. A summary of the various methods is presented in Table 1.

### 2.1. Transient Fluorescent Reporter Lines

A transient reporter line is produced by introducing the reporter construct as a non-integrating plasmid or vector, e.g., adenovirus [19] into the host cell (Figure 1A). The reporter expression level obtained is dependent upon the copy number, and typically decays as the non-integrated copies are lost, leading to cell-to-cell phenotypic variation. An application in developmental studies is limited since the transfection needs to be performed at a time where the ectopic promoter driving the FP is active. To avoid such difficulties, stable lines generated by the following methods are preferred. 

### 2.2. Random Integration of a Fluorescent Reporter Transgene

This method relies on the reporter construct being cloned, packaged and delivered via a genome-integrating viral system, e.g., a lentiviral vector [19] (Figure 1B) or in the form of linear or circular DNA. In this latter method, a plasmid or bacterial artificial chromosome (BAC) encoding the fluorescent reporter is introduced by chemical transfection or electroporation and the stability of the integration is ensured through antibiotic selection. 

This technique is relatively simple but possesses many biological and technical risks attributable to the random insertion into the genome [20]. Firstly, the transgene copy number cannot be controlled and the insertion of multiple copies of the construct has been linked to a loss of transgene expression in hESCs [21]. Secondly, the transgene expression may be influenced by the genomic context [22] and vice versa, leading to clonal variability in the fluorescent signal. Thirdly, transgene silencing upon differentiation may occur if the transgene is marked by epigenetic factors, or if the transgene is under the influence of altered chromosomal architecture [23,24,25]. All these risks lead to a situation where the fluorescent signal may not reliably mirror the GOI’s promoter activity, confusing the downstream analysis.

### 2.3. Targeted Integration of a Fluorescent Reporter Transgene

In order to avoid the undesirable effects of a random integration, fluorescent reporter constructs can be inserted into specific genomic sites. Homologous recombination (HR) (Figure 1C) [26] is a DNA repair process that utilises homologous DNA as a template to regenerate lost sequence. Transfecting cells with a reporter construct harbouring an FP flanked with upstream and downstream “arms” homologous to the target locus, the sequence in between the homology arms can be exchanged, and thus the FP inserted. The main disadvantage of this method lies in the low targeting efficiency that can make such experiments challenging [27]. 

### 2.4. Advanced Genome Engineering Technologies 

The development of site-specific endonucleases including zinc finger nucleases (ZFNs), transcription activator-like effector nucleases (TALENs) and clustered regularly interspaced short palindromic repeat (CRISPR)/CRISPR associated protein 9 (Cas9) has transformed our ability to manipulate the genome [28]. These custom-designed genome-engineering nucleases serve as molecular scissors to cleave specific DNA sequences. The introduction of a double strand break (DSB) can stimulate the HR mechanism, thereby increasing its efficiency [29] (Figure 1D).

With respect to site-specific integration of a fluorescent reporter, there are two common strategies. In the first strategy, the reporter construct can be introduced into a so-called safe harbour site (SHS) (Figure 1E), e.g., adeno-associated virus integration site 1 (*AAVS1*) [30], chemokine receptor 5 or the human homolog of murine *Gt(ROSA)26Sor* [31]. These genomic sites have been found to be suitable for transgene expression, ensuring the stable expression of the FP in the majority of cell types [32] and the insertion event is not associated with any major deleterious consequences. In this scenario, a reporter construct is flanked with homology arms designed against the SHS sequence. To simplify insertion at these well-used sites, systems have been engineered which enable a catalytic insertion of a transgene with recombinases. Here, recombination sites, such as heterotypic *LoxP* or *Frt* sites, are inserted into the SHS, and equivalent sites are incorporated into the reporter construct, enabling the sequences to be efficiently exchanged through the activity of recombinases [33]. This method, known as recombination-mediated cassette exchange (RMCE), demands the production of “a master cell line” into which multiple fluorescent reporter derivatives can be inserted, permitting the production of isogenic lines without the requirement for retargeting by HR [31,34]. 

A general disadvantage for all exogenous fluorescent reporter constructs is that a minimal promoter region is used. Subsequently, the fluorescence readout may not faithfully recapitulate the biological expression of the GOI. To avoid this issue, endogenous regulatory elements can be exploited for FP expression, allowing a higher fidelity between reporter and actual promoter activity. Using HR, FPs can be inserted downstream (Figure 1F) of an endogenous GOI or directly upstream of the endogenous start codon. This strategy can directly perturb the GOI locus leading to loss-of-function; hence, a careful design is a prerequisite to ascertain whether the normal function of the GOI is to be retained. Bicistronic expression of the FP downstream of the GOI may provide one way of avoiding loss-of-function. In these cases, linker sequences such as porcine teschovirus-1 2A (P2A), Thosea asigna virus 2A (T2A) [35] or internal ribosomal entry sites (IRES) [36] can be included to separate the GOI ORF and FP [35,37].

Each construct design has inherent advantages and disadvantages and the exact approach adopted should be tailored to the study. Random integration may be a sensible approach for studying gene function and pre-screening of biological effects. A master cell line is more suitable for lineage-specific labelling where multiple reporter lines are needed. Lastly, endogenous locus targeting is safer for projects that rely on the sensitivity of the reporter system. 

## 3. PSC-CM Reporter Lines: An Insight into Cardiac Biology

In vitro PSC-CM differentiation involves sequential stages of mesoderm induction, cardiac mesoderm determination, lineage specification and terminal differentiation [18]. WNT/β-catenin signalling plays stage-specific, biphasic roles in CM differentiation [38]. Canonical WNT promotes mesoderm differentiation but inhibits subsequent cardiac mesoderm specification. Despite a greater understanding of developmental pathways, pure CMs are not attainable from in vitro differentiation of PSCs. Fluorescent reporter cell lines, therefore, serve as a powerful tool for deciphering key transition stages essential for cardiogenesis and thus help improve in vitro differentiation protocols. A summary of applications of fluorescent PSC-CM reporter lines, including the steps of in vitro cardiac development, are illustrated in Figure 2 with additional details of published studies presented in Table 2.

In a developing mammalian embryo, the earliest cardiovascular progenitors are identified in gastrulation, where cells enter epithelial-mesenchymal transition and form mesendoderm. These progenitors express a mesodermal marker T-box transcription factor brachyury (*BRY*) which has been used to drive a *GFP* fluorescent reporter. In vitro differentiation of mouse ESCs (mESCs) revealed a multipotency of canonical WNT-activated *Bry*-GFP^+^ cells. Under cardiogenesis-promoting conditions, *Bry*-GFP^+^ cells with low expression of kinase insert domain receptor (Kdr^low^) exhibit cardiogenic potential whilst, *Bry*-GFP^+^, Kdr^high^ populations possess haematopoietic potential [39,40]. 

The transition from common mesoderm towards cardiac mesoderm requires canonical WNT inhibition and non-canonical WNT activation. These intermediate cells are marked by an expression of mesoderm posterior basic helix-loop-helix transcription factor 1 (*MESP1*). Investigations which used a fluorescently-tagged *MESP1-mCherry*/ NK2 homeobox 5 (*NKX2.5)*-*eGFP* dual reporter hESCs [41] demonstrated a transient *MESP1-mCherry* expression followed by *NKX2.5-eGFP* expression. Enrichment of *WNT5A*, receptor tyrosine kinase-like orphan receptor 2 and frizzled class receptor 2 in *MESP1*-mCherry^+^ cells suggested roles of non-canonical WNT pathway in cardiac progenitor determination [42].

The nascent cardiac progenitors undergo another transitory stage towards more defined cardiac lineages termed the first heart field (FHF), which contributes to the left ventricle and a portion of the atria, and the second heart field (SHF), which gives rise to the right ventricle, atria, outflow tract and inflow tract [43]. In the human heart, insulin gene enhancer protein (*ISL1*) is a potential SHF marker [44]. A study using an *ISL1-cre dsRed* reporter hESC line [44] ascertained an ISL1^+^ multipotent status with differentiation potential for CMs, smooth muscle, endothelial and epicardial lineages. This finding may highlight a common ancestor shared between pro-epicardial progenitor and SHF CPCs since time-course analysis showed that the *ISL1* expression preceded the expression of SHF markers T-box transcription factor 1 and *NKX2.5*. *NKX2.5/ISL1* or *ISL1*/epicardial-specific Wilm’s tumour protein 1 dual reporter systems may be able to elucidate the diversification of ISL1^+^ progenitor and SHF CPCs further. More recently, CRISPR-Cas9-assisted generation of a *NKX2.5-TagRFP*/T-box transcription factor 5 (*TBX5)*-*Clover* dual reporter hiPSC line [45] has offered a phenotypic strategy to isolate four distinct subsets of cardiac progenitors (CPCs); presumptive FHF (*NKX2.5*-TagRFP^+^/*TBX5*-Clover^+^); presumptive SHF (the *NKX2.5*-TagRFP^+^/*TBX5*-Clover^−^); pro-epicardial cells (*NKX2.5*-TagRFP^−^/*TBX5*-Clover^+^), which diversified into nodal-like CMs and epicardial cells; and endothelial progenitors (*NKX2.5*-TagRFP^−^/*TBX5*-Clover^−^) (Figure 3).

Taken together, the stage-specific fluorescent PSC-CM reporter cell lines offer an optical tracking strategy for cell identity and enable in vitro studies of cardiac development. Uniquely, hiPSC-CM reporter cells provide insights into human cardiogenesis without embryo donor requirements and without the concerns over species differences encountered with mouse models. Future multi-coloured reporter lines [46] may enable the discovery of novel cardiac progenitors, and the improvement of CM derivation and differentiation protocols.

## 4. Fluorescence-Guided PSC-CM Purification

Despite the refinement of in vitro cardiac induction protocols [67], PSC-CM purification is still needed to ensure sufficiently pure populations for downstream applications. Contamination by non-CM cell types, including fibroblasts, endothelial, epicardial and smooth muscle cells [45,68,69], and undifferentiated cells may hamper the detection and analysis of the actual signals from CMs. The heterogeneity of PSC-CM subtypes may also dilute the degree of pharmacological response originated from a certain subtype.

Percoll gradient centrifugation [70] allowed hES-derived CMs enrichment up to 70% from initial beating embryoid bodies, but this technique is labour-intensive and requires high numbers of large beating embryoid bodies. Metabolic selection [71] enables the elimination of non-cardiac cells that rely on the glycolytic pathway, although may be subject to biases which could lead to considerable variability in CM yield due to the metabolic immaturity of PSC-CMs [72]. Relying on the high mitochondrial content in CMs, fluorescent mitochondrial dyes offer a way in which CMs can be transiently labelled and purified by fluorescence-activated cell sorting (FACS) [73]. Nonetheless, the purification outcome will depend on the stage of PSC-CM maturity [74].

Genetically engineered reporter lines provide a robust alternative. Reporter lines carrying an antibiotic-resistant cassette under the control of cardiac-specific gene’s regulatory elements, e.g., α-myosin heavy chain (*MYH6*) [75,76,77,78,79,80], cardiac troponin (*TNNT2*) [51], *NKX2.5* [68] and Na^+^/Ca^2+^ antiporter (*NCX1*) [53] allow effective PSC-CM purification (up to 99% purity) [75,76]. However, the consequence of antibiotic selection may adversely influence the native physiology of the cells. Prolonged drug application, e.g., puromycin and neomycin in CMs was reported to modulate Ca^2+^ transient (CT) reduction and action potential (AP) shortening [80,81]. Meanwhile, fluorescent reporter cell lines not only allow PSC-CM purification via FACS, but also enable real-time tracking of the cells during differentiation and maturation. As we explore below, the choice of control elements incorporated in the fluorescent reporter will vary widely depending on the target population to be distinguished.

### 4.1. Functional CMs

*NKX2.5* is an early transcription factor in the FHF and SHF CPCs and plays supportive roles in cardiac contraction [82]. An engineered *NKX2.5-GFP* reporter hESC line [47] allowed efficient functional CM purification, as confirmed by electrophysiological phenotype. It is should be noted that *NKX2.5* expression is also detectable in non-cardiac tissues such as the spleen, stomach and thyroid [83]; hence, exploiting multiple markers in addition to *NKX2.5* may be advantageous.

Fluorescent reporter transgenes under the control of a sarcomere gene/promoter offer a more cardiac-specific cell labelling. Isolated fluorescent *MYH6-*GFP^+^[48,49] or *MYH6*-mCherry^+^ [50] hPSC-CMs expressed pan-cardiac markers while each cell displayed functional phenotypes representing atrial, ventricular or nodal CMs. Importantly, as CMs mature, the cells undergo sarcomere reorganisation, including myosin heavy chain α-to-β isoform switching in human or β-to-α switching in mouse [84,85,86]. Therefore, *MYH6* reporter systems are better suited for the isolation of early committed human CMs and vice versa for mouse CMs. The ability of sarcomere-specific reporters to aid in isolation of beating CMs was also recently reported for a number of key genes: α-cardiac actin (*ACTC*)-*mCherry* [51], *TNNT2*-firefly luciferase */*puromycin resistance [51] and titin (*TTN*)-*GFP* hPSC lines [52]. In addition to the CM purification utility, the fluorescent N-terminal tagging of endogenous titin facilitated observation of the sarcomere dynamics in a cardiomyopathy condition [87].

*NCX1* plays essential roles in intracellular Ca^2+^ flux and excitation-contraction coupling [54]. An *NCX1*-*eGFP* hPSC reporter line [53] offered an optical selection method for functional PSC-CMs. Although the fluorescent signals preceded the contractile activity, the sorted *NCX1*-eGFP^+^ cells displayed both molecular and functional characteristics of typical hPSC-CMs.

Irrespective of the pan-cardiac promoter used in these reporter systems, FACS purification yielded a mixture of atrial, ventricular and nodal CM subtypes. These reporter lines benefit cardiotoxic screening and developmental studies, but the heterogeneous nature of the cell population complicates chamber-specific drug discovery, disease modelling and therapeutic applications. Each subtype exhibits unique physiological characteristics [88], leading to different drug responses. Similarly, some pathological conditions, e.g., atrial fibrillation, develop within specific heart regions, thus purified CM subtypes serves as a more precise disease model [89]. Furthermore, subtype mismatching or mislocation may exert adverse effects in cardiac cell therapy, such as pro-arrhythmia. Therefore, CM subtype selection is necessary to overcome these limitations.

### 4.2. Ventricular CM

Myosin light chain 2 (*MYL2*), is the most attractive ventricular marker with an expression restricted to ventricular compartments throughout mammalian development [85,90]. Success in ventricular-like CMs enrichment by FACS has been reported in several fluorescent *MYL2* PSC-CM reporter lines [54,55,56,57,58], each of which was verified by molecular and/or electrophysiological phenotype. It is noteworthy that *MYL2* is considered a late cardiac gene, where its expression level increases consistently with the stage of CM maturation, with the upregulation seen being more pronounced than other maturation factors, such as β myosin heavy chain [91]. Therefore, the selection of a suitable time point for sorting *MYL2*-FP+ cells is important to optimise the yield of ventricular-like cells.

### 4.3. Atrial CM

The selection of atrial-like cells is more challenging owing to the lack of a reliable genetic marker. Myosin light chain 7 (*MYL7*) is ubiquitously expressed within the primitive heart tube and only becomes confined to atrial compartments at later developmental stages [92]. Analysis of a transient *MYL7-GFP* hiPSC-CMs identified co-expression of *MYL2* and *MYL7* in 96% of the *MYL7*-GFP^+^ fraction [57], perhaps underscoring the foetal nature of PSC-CMs. Nevertheless, stable reporter lines, especially those established by gene targeting approaches, may behave differently.

During mammalian heart development, sarcolipin (*SLN*) expression is confined to atrial lineages [93]. Electrophysiological measurements revealed atrial characteristics of sorted *SLN*-tdTomato^+^ hiPSC-CMs [59], highlighting *SLN* as an alternative atrial marker. Following this report, an *SLN*-driven voltage-sensitive fluorescent protein (*SLN-VSFP*) hiPSC reporter line was generated [58], which allowed FACS isolation of atrial-like CMs coupled with the functional study of the sorted cells by optical AP imaging. However, the temporal concern of *SLN*-based atrial cell purification has been emphasised due to the downregulation of the gene as the cells become more mature [59].

Evidence from a gene knockdown in hiPSC-CMs, supported by human tissue analysis, suggested nuclear receptor *NR2F1/2* as being important players in retinoic acid-induced atrialisation [94]. This was exploited with an *NKX2.5-GFP/NR2F2-mCherry* hESC dual reporter system, enabling two chamber-type CMs to be distinguished [60]. Molecular and electrophysiological analysis of *NKX2.5*-GFP^+^/*NR2F2*-mCherry^+^ confirmed an atrial phenotype while *NKX2.5*-GFP^+^/*NR2F2*-mCherry^-^ cells showed more ventricular characteristics.

### 4.4. Nodal CM

PSC-derived nodal-like CM enrichment is demanding due to this subtype being a relatively minor population under WNT-modulated differentiation procedures [95]. Investigations in transgenic mice found chick *GATA6* (*cGATA6*) enhancer activity restricted to the atrioventricular conduction system [96]. An engineered *cGATA6-eGFP* hESC reporter line [61] enabled FACS enrichment of non-working CMs (95% purity) with expected nodal type APs and expression of nodal-specific hyperpolisation-activated cyclic nucleotide-gated K+ channel 4 (*HCN4*).

A transgenic short stature homeobox 2 (*SHOX2*)-*VSFP* hiPSC line [58] offered an alternative genetic selection of nodal-like cell isolation, as confirmed by high expression of *HCN4* and nodal-type AP. *SHOX2* promotes the transcription of *HCN4* [97], the key channel responsible for the pacemaker-specific funny current. The transient expression of exogenous *SHOX2* was sufficient to induce endogenous *SHOX2* expression and steered the differentiation of ESCs towards the nodal lineage [98].

In summary, fluorescent PSC-CM reporter systems facilitate optical CM identification and, depending on the cardiac-specific control element used, isolation of sub-populations from the pool of PSC-derived cells. Exploiting pan-cardiac promoters allows purification of mixed atrial, ventricular and nodal CMs while avoiding non-CMs contamination, providing a source for in vitro cardiotoxic screening and developmental studies. Meanwhile, each CM subtype can be optically selected using subtype-specific fluorescent PSC-CMs reporter lines. The development of CM subtype purification techniques holds great promise for CM subtype-selective drug discovery, subtype-specific disease modelling and scalable subtype-specific CM production that future cardiac regeneration therapies may demand.

## 5. Applications towards Optical Measurements of Cardiac Functions

Electrophysiological measurements are not only a reliable strategy to identify individual CM subtypes, they also allow investigation of either genetically inherited or acquired functional cardiac defects. Patch-clamp electrophysiology is the gold standard AP recording technique [99], but it is labour-intensive and low throughput. Intracellular Ca^2+^ level coordinates CM excitability and contraction. Membrane depolarisation causes Ca^2+^ influx through the L-type voltage-gated Ca^2+^ channel, in addition to NCX1 activity [54], which triggers Ca^2+^-induced Ca^2+^ release from the sarcoplasmic reticulum, thereby activating the contraction [100]. Although the use of fluorescent Ca^2+^ dyes (such as Fura2/4) for CT measurement is common, these chemicals are known to suppress cardiac functions [101] and prevent long-term studies of pharmacological effects [102]. Genetically encoded voltage and Ca^2+^ reporters are being explored as we outline below.

### 5.1. Genetically Encoded Fluorescent Voltage-Sensitive and Ca^2+^ Indicators

The development of genetically encoded fluorescent voltage-sensitive indicators (GEVIs) has opened a new window for electrophysiological studies. Arclight is a synthetic protein containing a voltage-sensing domain from *Ciona intestinalis* and a super ecliptic pH-sensitive GFP variant (pHluorin A227D) [103,104]. Randomly integrating a *CAG-Arclight* transgene into hiPSC-CMs enabled the ubiquitous expression of the GEVI driven by a synthetic *CAG* control element [105]. The fluorescence pattern observed in these cells was comparable to the AP recording using fluorescent Di-8-ANEPPS dye and even recapitulated drug-induced and genetically-caused AP prolongation and arrhythmias (long QT2 syndrome) [62,106]. Furthermore, the *Arclight* reporter showed improved signal-to-noise ratios and was less affected by photobleaching than conventional dyes. To facilitate simultaneous subtype identification and electrophysiological studies, a GEVI can be directed by a CM subtype-specific promoter. As an alternative, VSFP-based GEVIs [58] rely on Förster resonance energy transfer (FRET) [107] to track oscillations of membrane potential causing conformational changes of two attached tandem FPs, resulting in the dynamic rise and fall of the fluorescent intensity of FRET donor and FRET receptor. The ratiometric readout of VSFPs, compared to the single fluorescent unit used in the *Arclight* system, provides a more stable AP pattern, unaffected by movement artefacts and photobleaching.

Several versions of genetically encoded fluorescent Ca^2+^ indicators (GECIs) have been engineered and tested in PSC-CMs [62,63,65,106]. GECIs contains three major components: a Ca^2+^-modulating protein (CaM), a CaM-interacting peptide, e.g., myosin light chain kinase-derived M13 and a circularly permuted GFP [108]. Following Ca^2+^-induced Ca^2+^ release, cytosolic Ca^2+^ binds to CaM which promotes the formation of Ca^2+^-CaM-M13, resulting in increased fluorescence [109]. Similar to GEVIs, the key advantage of GECIs over traditional fluorescent Ca^2+^ dyes is the stable expression of the transgene, facilitating observations of the chronic effects of pharmacological molecules, and non-cytotoxic properties [63]. The fidelity of GECI reporter systems is comparable to fluorescent dyes, with abilities to elucidate drug-induced changes in beat rate and beat interval. Furthermore, such genetic reporters have been used to model inherited arrhythmic disorders such as catecholaminergic polymorphic ventricular tachycardia type 2 [62,106].

### 5.2. Optogenetics

The field of optogenetics has captured interest in neurophysiology because of the ability to track spatiotemporal activation of neurons [110]. Upon blue light illumination, the *Chlamydomonas reinhardtii*-derived channelrhodopsin 2 protein (ChR2) mediates inward cation currents [111], resulting in membrane depolarisation. In the PSC-CMs field, optogenetics enables light-triggered synchronisation of beating CM patches and promotes CM maturation [112]. Differentiated PSC-CMs contract spontaneously but largely non-uniformly, potentially resulting in AP and CT variance. Instead of using an invasive electrical impulse, *CAG*-humanised *ChR2* (*hChR2*)-*mCherry* reporter hiPSC-CMs could be optically paced at different beat rates [64], offering an in vitro model to study rate-sensitive conditions, e.g., long QT1 [113]. Furthermore, a combination of the optogenetics technology with GECIs and/or GEVIs [63] allows an all-optical detection for high throughput drug screening purposes.

## 6. Emerging Insights into Cardiac Tissue Repair

The adult human heart has limited regenerative capacity to self-repair [114]. hiPSC-CMs have been proposed as a promising approach for immunocompetent cardiac cell therapy. As a vision for future treatment, somatic cells, e.g., skin fibroblasts or blood cells, could be extracted from a patient and reprogrammed into iPSCs. In vitro cardiac differentiation of these cells would yield an autologous CM source, allowing reintroduction without immune-rejection [115]. Nonetheless, the immature nature of the potential hiPSC-CM graft raises concerns over clinical safety [112,116].

The proliferative capacity of stem cells is desirable for cardiac regeneration to compensate for the massive cell loss seen following transplantation [117]. However, subcutaneous injection of purified mouse iPSCs (miPSCs) ubiquitously expressing a tri-reporter into mice suggested a significant tumourigenic potential by bioluminescent signal, as compared to miPSC-CMs [66]. This indicates that iPSC-derived CPCs may be a safer source of cells for cardiac repair [118]. In support, biopsy analysis of the pre-infarcted mouse heart intramyocardially transplanted with purified *Nkx2.5*-GFP^+^ miPSCs confirmed the cardiac differentiation potential of the CPCs without teratoma formation [119]. Future multi-coloured reporter CPC lines will offer a wide variety of transplants customisable to specific impaired region.

Host-graft adaptation is another concern for cardiac cell therapy. Importantly, fluorescent PSC-CM reporters provide not only a tool for tracking the cell fate after transplantation but also provide information about graft–host structural and functional integration. Histological analysis of rat heart pre-intramyocardially injected with *MYL2*-eGFP^+^ hPSC-CMs revealed signs of morphological maturation and connexin 43 (*CX43*) expression at the host–graft interface [55]. Since CX43 is a major ventricular gap junction protein that plays an integral role in the excitation–contraction coupling [120], the presence of the donor CX43 at the donor–recipient cell interface may be indirectly indicative of functional adaptation of the graft.

Notwithstanding, the immaturity of hPSC-CMs still represents a challenge for cardiac cell therapy. New methodologies focused on promoting more adult phenotypes are needed to ensure clinical safety and maximize therapeutic benefit.

## 7. Maturation Approaches

Several techniques have been developed to improve iPSC-CM maturity including biochemical, physical and environmental stimulation [121]. Modulating pathways promoting CM differentiation improves adult CM phenotypes. For example, insulin-like growth factor/AKT1 supplementation supports the expression of CM markers and enhances intracellular Ca^2+^ fluxes and β-adrenergic response [122]. Similarly, thyroid hormone increases hiPSC-CM contractile force, Ca^2+^ cycling and mitochondrial respiratory capacity while promoting sarcomere organisation [123].

Unlike PSC-CMs, primary adult atrial and ventricular CMs do not beat spontaneously, but require an electrical signal from nodal CMs. Mimicking the in vivo situation, extrinsic electrical or optical (for optogenetics) impulses not only help to pace cultured CMs but also exert positive effects on the ultrastructural organisation and functional properties [124,125].

Extracellular matrix stiffness changes dramatically during cardiac development and has an impact on the mechanical force experienced by CMs [126]. 3-dimentional (3D) culture provides closer phenotypes to native CMs than conventional 2-dimentional culture, facilitating in vivo topology that supports cell–cell interactions and exposure to biochemical and physical factors. Advances in the fields of biomaterial [127] and 3D PSC-CM culture present an opportunity to attain more mature PSC-CMs. Engineered heart tissue [128] and cardiac microtissue [129] are generated by casting a cell–hydrogel mixture in moulds tethered to elastic anchors. The mechanical force applied on the anchors (or a passive stretcher for ring-shaped engineered heart tissues [130]) can be varied to simulate physiological/pathophysiological conditions.

Combinations of 3D hiPSC-CMs with biochemical [106] or electrical [131] administrations promote adult phenotypes and enhance electrophysiological properties and contraction [132]. The presence of non-CMs such as fibroblasts [69,133], endothelial [133] and epicardial cells [134] also elicits favourable consequences on PSC-CM differentiation and maturation in 3D format. GECIs [62], GEVIs [62] and optogentics [135] have been applied to 3D hiPSC-CM culture recently, which allow optical measurements of electrophysiology in normal and drug-induced or inherited diseased models. Meanwhile, a fluorescent reporter that is tagged to a maturation marker, e.g., *CX43* [85] may be beneficial to follow the maturation of implants. Similarly, the transduction of GECI and GEVI into PSC-CMs would allow in vivo studies for functional adaption/maturation of the graft.

## 8. Conclusions

The indefinite source of PSC-CMs has the potential to provide scalable CM production for cardiovascular research. However, immature and embryonic-like phenotypes as well as the heterogeneous nature of PCS-CMs present challenges for their use in drug discovery, disease modelling and transplantation. Stage-specific fluorescent PSC-CM lines generated by different transgenic methods have helped researchers monitor cardiac development and improved in vitro differentiation protocols. The discovery of additional CM maturation-promoting pathways will require multi-coloured reporter systems in combination with advances in 3D PSC-CM culture.

CM surface markers are largely unknown and thus FACS-mediated PSC-CMs sorting by fluorescent markers provide a genetic-based solution for specific cell-type isolation. Purified fluorescent PSC-derived CPCs have been successfully differentiated into functional CMs in vivo, devoid of tumorigenic risk, promising clinical safety for cardiac regeneration. Novel fluorescent PSC reporter lines will help identify the various CPCs that give rise to unique cell-type compositions specifically required for treating different pathological conditions. Furthermore, the isolation of subtype-specific PSC-CMs will enable subtype-specific cardiogenic drug discovery and disease modelling. Unlike nodal and ventricular types, the atrial PSC-CM markers are not fully characterised. Further insights from novel fluorescent atrial PSC-CM reporter lines, such as *PITX2* and *KCNA5*, in combination with atrial-promoting differentiation approaches including retinoic acid treatment and CRISPRa/CRISPRi-mediated transcriptional activation/inhibition of atrial-regulating pathways [136,137], could assist the development of atrial cell isolation protocols.

Lastly, optogenetics is an emerging biotechnology that has recently been applied in PSC-CMs for spatiotemporal CM activation and maturation. The combination of optogenetics with fluorescent physiological reporters allows an all-optical measurement of CM function that benefits high throughput drug screening. We look forward to seeing the implementation of combined optogenetics and physiological sensors with minimised optical crosstalk in cell transplantation, which will allow PSC-CM graft performance to be followed in vivo.

## Figures and Tables

**Figure 1 biology-09-00402-f001:**
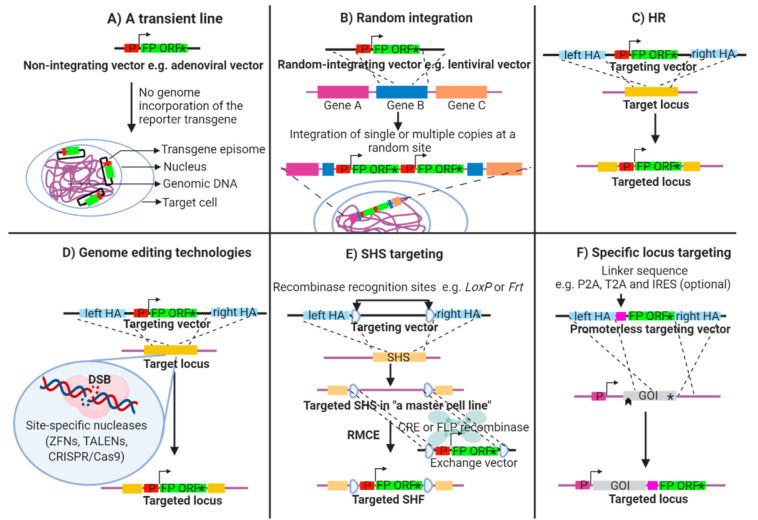
Schematic illustration of transgenic methods for fluorescent reporter line generation. A simple fluorescent reporter vector consists of an ectopic fluorescent protein (FP) open reading frame (ORF) preceded by a promoter (P) of a gene of interest (GOI) (**A**,**B**). (**A**) A non-integrating vector, e.g., adenoviral vector is used to produce a transient reporter line whereas using (**B**) a random-integrating vector, e.g., lentiviral vector results in a genomic insertion of either a single copy or a concatemer of the transgene at random loci. (**C**) Positioning a reporter construct at a specific locus via homologous recombination (HR), flanking genetic arms (HA) homologous to a target site are required. (**D**) The HR efficiency can be improved by incorporating genome editing technologies, e.g., zinc finger nucleases (ZNFs), transcription activator-like effector nucleases (TALENs) and clustered regularly interspaced palindromic repeat (CRISPR)/CRISPR-associated protein 9 (Cas9), which introduce a double strand break (DSB) at a specific site, stimulating HR. A fluorescent reporter can be integrated into (**E**) a safe harbor site (SHS) or (**F**) an endogenous GOI. (**E**) “A master cell line” can be prepared by specifically introducing a pair of recombinase recognition site, e.g., *LoxP* and *Frt* at a SHS. Subsequently, the flanked sequence can be exchanged for a fluorescent reporter construct by transfecting an exchange vector carrying a reporter construct flanked by equivalent recombination recognition sequences together with a corresponding recombinase enzyme, e.g., CRE or FLP, facilitating the production of multiple fluorescent reporter lines from the same master line via a method known as recombinase-mediated cassette exchanged (RMCE). (**F**) To allow FP expression under an endogenous gene promoter, a promoterless targeting vector is used to target a specific GOI, e.g., at the stop codon. Alternatively, a link sequence, e.g., P2A, T2A and IRES is added between the FP and the target gene ORF to allow bicistronic expression of both the fluorophore and the target protein.

**Figure 2 biology-09-00402-f002:**
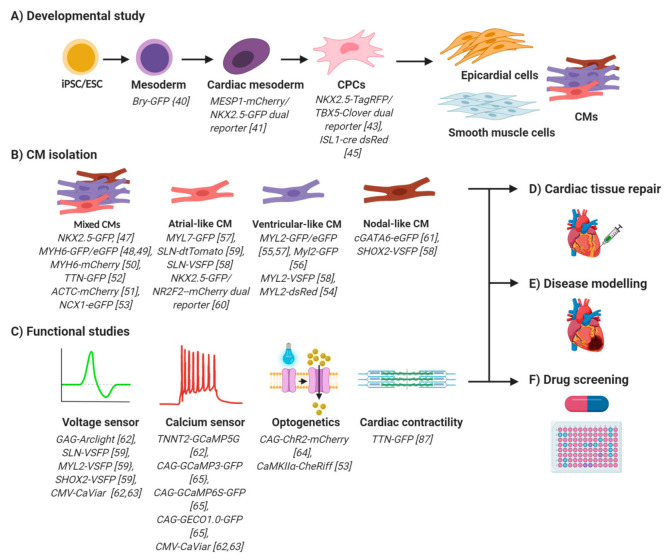
An overview of potential applications for fluorescent PSC-CMs reporter lines. (**A**) Stage-specific fluorescent reporters provide insights into cardiac development, which help improve PSC-CM differentiation protocols. (**B**) CM-specific PSC-CM fluorescent reporter lines facilitate FACS-based CM isolation. Purification that is based on a pan-cardiac promoter activity yields mixed types of CMs, each of which can be further selected by using subtype-specific PSC-CM reporter lines. (**C**) Optical action potential and Ca^2+^ dynamic measurements can be made using PSC-CMs lines expressing fluorescent voltage (first panel) and fluorescent Ca^2+^ sensors (second panel). Optogenetic technology aids in spatiotemporal CM activation, which can be used to pace CMs optically for rate-sensitive electrophysiology (third panel). Fluorescent tagging of the sarcomere components such as titin allows a real time observation of CM contractility (fourth panel). Exploiting the advantages of fluorescent PSC-CM reporter lines for cell purification and functional investigation facilitate (**D**) cardiac tissue repair, (**E**) disease modelling and (**F**) high throughput drug screening.

**Figure 3 biology-09-00402-f003:**
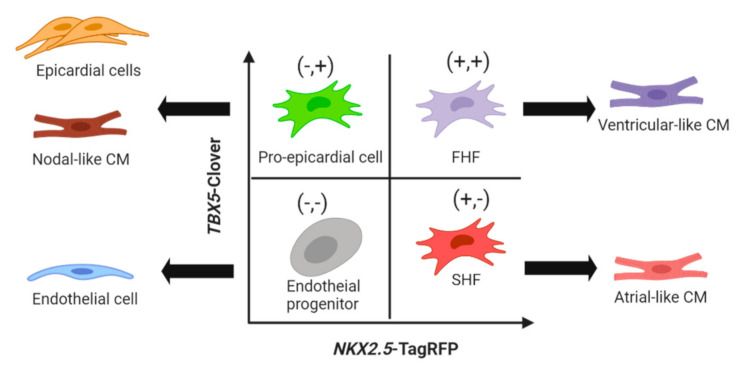
The *TBX5-Clover/NKX2.5-TagRFP* reporter PSC-CM line uncovers four distinct cardiac progenitor subsets [45]. The *TBX5*-Clover^+^/*NKX2.5*-TagRFP^+^ (+,+) (**top right**) and *TBX5*-Clover^−^/*NKX2.5*-TagRFP^+^ (+,−) (**bottom right**) show phenotypic features of presumptive first heart field and second heart field progenitors which give rise primarily to ventricular-like and atrial-like CMs, respectively. The *TBX5*-Clover^+^/*NKX2.5*-TagRFP^−^ (+/−) population (**top left**) resembles epicardial precursors which can then differentiate into epicardial cell and nodal CM lineages. The *TBX5*-Clover^−^/*NKX2.5*-TagRFP^−^ (−,−) population (**bottom left**) represents distinct endothelial lineage progenitors.

**Table 1 biology-09-00402-t001:** A summary of different transgenic approaches used to generate fluorescent reporter cell lines.

Method	Advantage	Disadvantage
(1) Transient fluorescent reporter lines	Simple and easy preparation/cloning of transgene donor High expression of an FP immediately after transfection	Rapid loss of FP expression after transfection Uncontrolled copy number of transgene
(2) Random integration of a fluorescent reporter transgene	Simple and easy preparation/cloning of transgene donor Very efficient method to generate reporter cell line	Uncontrolled copy number of transgene Risk of transgene silencing upon differentiation Risk of regulatory interference of nearby genes
(3) Targeted integration of a fluorescent reporter transgene	Controlled copy number of transgene	Multiple steps of targeting vector generation
(3.1) Homologous recombination	No additional nuclease requirement	Low efficiency of gene targeting
(3.2) Advanced genome engineering technologies	Improved targeting efficiency Commercially available source of nucleases (Cas9) and targeting design platforms	Off-target effects Laborious protein engineering steps (for TALENs and ZFNs)
(3.2.1) Safe habour site targeting	Low risk of transgene silencing Commercially available targeting vector especially for *AAVS1* locus Amendable system for multiple reporters via recombinase-mediated cassette exchange	Uncertain biological readout–exogenous promoter activity may not recapitulate the endogenous status
(3.2.2) Specific locus of interest targeting	Faithful biological readout Multiple reporters in the same cell (multi-coloured reporter system)	Locus-dependent targeting efficiency variation Complicated multiple line generation–each locus requires a unique targeting design Risk of destabilizing expression of endogenous gene

**Table 2 biology-09-00402-t002:** A summary of published fluorescent PSC-CM reporter systems.

Reporter Construct	Type of Promoter Used	Labelled Cell Population	Utility	Host PSC System	Transgenesis Method	Ref
*Bry-GFP*	Endogenous *Bry*	Mesoderm	Tracking mesodermal cell differentiation towards cardiogenic and haematopoietic lineages	mESC	HR by electroporation of the BAC-derived targeting vector into mESCs	[40]
*MESP1-mCherry*/*NKX2.5-eGFP*	Endogenous *MESP1* and *NKX2.5*	Cardiac mesoderm (*MESP1*) and CPCs (*NKX2.5*)	Tracking of transition of cardiac mesoderm toward CPCs and subsequent CM differentiation	hESC	HR by electroporation of BAC-derived targeting vector into hESCs	[41]
*NKX2.5*-IRES- *TagRed*/*TBX5*-IRES-*Clover*	Endogenous *NXK2.5* and *TBX5*	CPC subsets	Identification and characterisation of presumptive FHF, SHF, pro-pericardial and pre-endothelial progenitors	hiPSC	CRISPR/Cas9 editing in hiPSC at the *NKX2.5* and *TBX5* loci using plasmid-based lipofection	[45]
*ISL1-cre dsRed*	Endogenous *ISL1*	ISL1^+^ CPC	Characterisation of ISL1^+^ CPC derivation	hESC	HR by electroporation of BAC-derived *ISL1*-*cre* vector into hESCs followed by transfection of floxed dsRed vector	[44]
*NKX2.5-GFP*	Endogenous *NKX2.5*	NKX2.5^+^ CPC/CMs	Isolation of NKX2.5^+^ CPC and CMs	hESC	HR by electroporation of BAC-derived targeting vector	[47]
*MYH6-GFP*	Ectopic *MYH6*	Beating CMs	Purification of beating CMs and development of PSC-CM differentiation protocol	hESCs and hiPSCs	Lentiviral transduction/ random integration of the transgene into differentiating hPSCs	[48]
*MYH6-eGFP*	Ectopic *MYH6*	CM progenitor /beating CMs	Identification of early CM progenitor and purification of CMs	hESC	Lentiviral transduction/random integration of the transgene into hESCs	[49]
*MYH6-mCherry*	Endogenous *MYH6*	Beating CMs	Purification of beating CMs for cardiotoxicity evaluation	hESC	CRISPR/Cas9 editing in hiPSC at the *MYH6* locus using plasmid-based nucleofection	[50]
*ACTC-mCherry*-WPRE-*EF1*-*neo*^¥^	Ectopic *ACTC*	Beating CMs	Purification of CMs	hiPSC	Lentiviral transduction/random integration of the transgene into hiPSCs	[51]
*TTN-GFP*	Endogenous *TTN*	Beating CMs	Purification of beating CMs and, potentially, study of sarcomere functions	hiPSC	CRISPR/Cas9 editing in hiPSC at the *TTN* locus	[52]
*NCX1-eGFP*-WPRE	Ectopic *NCX1*	Beating CMs	Purification of beating CMs	hiPSC and hESC	Lentiviral transduction/random integration of the transgene into hPSCs	[53]
*MYL2-dsRed*	Ectopic *MYL2*	Ventricular-like CMs	Purification of ventricular-like CMs	hESC	Lentiviral transduction/ random integration of the transgene into hESCs	[54]
*MYL2-eGFP*	Ectopic *MYL2*	Ventricular-like CMs	Purification of ventricular-like CMs	hESC	Lentriviral transduction/ random integration of the transgene into hESCs	[55]
*Myl2-eGFP*	Ectopic *Myl2* promoter + human *CMV*_ehc_	Ventricular-like CMs	Purification of ventricular-like CMs for cell transplantation study	mES	Random integration of the transgene into mESCs via electroporation delivery	[56]
*MYL2-GFP* ^φ^	Ectopic *MYL2*	Ventricular-like CMs	Purification of ventricular-like CMs	hiPSC	Adenoviral transduction of the transgene into hiPSCs	[57]
*MYL2-VSFP*	Ectopic *MYL2* promoter-enhancer	Ventricular-like CMs	Purification of ventricular-like CMs and study of optical AP	hiPSC	Lentiviral transduction/random integration of the transgene into hiPSC	[58]
*MYL7-GFP* ^φ^	Ectopic *MYL7*	Atrial-like CMs	Identification of atrial-like CMs	hiPSC	Adenoviral transduction/random integration of the transgene into hiPSC	[57]
*SLN-tdTomato*	Endogenous *SLN*	Atrial-like CMs	Purification of atrial-like CMs	hiPSC	HR by electroporation of recombineered BAC DNA into hiPSCs	[59]
*SLN-VSFP*	Ectopic *SLN*	Atrial-like CMs	Purification of atrial-like CMs and study of optical AP	hiPSC	Lentiviral transduction/random integration of the transgene into hiPSC	[58]
*NKX2.5-GFP/NR2F2-mCherry*	Endogenous *NR2F2* and *NKX2.5*	Atrial-like CMs	Identification and purification of atrial-like cells and functional study of *NR2F2*	hESC	CRISPR/Cas9 targeting (*NR2F2-mCherry*) into *NKX2.5-GFP* hESC reporter line [47]	[60]
*cGATA6-eGFP*	Ectopic *cGATA6* promoter-enhancer	Nodal-like cells	Identification and purification of nodal-like cells and development of PSC-derived nodal cell differentiation protocol	hESC	Lentriviral transduction/ random integration of the transgene into differentiated hESC-CMs	[61]
*SHOX2-VSFP*	Ectopic *SHOX2*	Nodal-like cells	Purification of nodal-like CMs and study of optical AP	hiPSC	Lentiviral transduction/random integration of the transgene into hiPSC	[58]
*CAG-Arclight*	Synthetic *CAG*	Ubiquitous	Overexpression of *Arclight* GEVI for optical AP recording in normal and pathological condition in engineered heart tissue	hiPSC	Lentiviral transduction/random integration of the transgene into healthy individual derived hiPSCs and long QT2 patient-derived hiPSCs	[62]
*TNNT2-GCaMP5G*	Ectopic *TNNT2*	Beating CMs	Overexpression of *GCAPMP5* GECI for optical CT recording in normal and pathological condition in engineered heart tissue	hiPSC	Lentiviral transduction/random integration of the transgene into healthy individual derived hiPSCs and catecholaminergic polymorphic ventricular tachycardia type 2 patient-derived hiPSCs	[62]
*CaViar* (*CMV-Arch(D95N*)-linker-*GCaMP5G*-WPRE)	Ectopic *CMV* promoter-enhancer	Ubiquitous	Overexpression of *Arch*(D95N) GEVI and *GCaPMP5* GECI for simultaneous recordings of AP and CT	hiPSC	Lentiviral transduction/random integration of the transgene into hiPSC	[62]
*CaViar* (*CMV-QuasAr2*-linker-*GCaMP6f*-WPRE) OR *CaMKIIa*-*Cheriff-eGFP*	Ectopic *CMV* promoter-enhancer OR ectopic Ca^2+/^calmodulin-dependent protein kinase II (*CaMKIIa*) promoter	Ubiquitous	Overexpression of *QuasAr2* GEVI and *GCaMP5f* GECI for simultaneous optical AP and CT recordings in combination with overexpression of optogenetic *Cheriff* for synchronising CM contraction	hiPSC	Lentiviral transduction/random integration of the transgene into differentiated hiPSC-CMs	[63]
*CAG-hChR2*-*mCherry*^φ^	Synthetic *CAG*	Ubiquitous	Transient overexpression of the optogenetic *ChR2* for frequency-dependent drug screening purpose	hiPSC	Adenoviral transduction of the transgene vector into differentiated hiPSC-CMs	[64]
*CAG-GCaMP3* OR *CAG-GCaMP6s* OR *GAG-G-GECO1.0*	Synthetic *CAG*	Ubiquitous	Overexpression of a different GECI version for optimising optical CT imaging in rhesus iPSC-CMs	Rhesus iPSC	CRISPR/Cas9 editing in rhiPSC at the *AAVS1* locus via plasmid-based chemical transfection	[65]
*CMV*-tri reporter	Ectopic *CMV*	Ubiquitous	Overexpression of luciferase (for bioluminescent cell tracking after transplantation), *mRFP* (purification of transduced cells) and thymidine kinase (following dividing cells and tumouriginicity)	miPSC	Lentiviral transduction/random integration of the transgene into miPSC	[66]

^¥^; WPRE, Woodchuck hepatitis virus posttranscriptional regulatory element used for enhancing gene expression; *EF1α*, eukaryotic translation elongation factor 1 α promoter driving neomycin resistant gene expression (*neo*). ^φ^; transient lines.

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
