# Peer review of "Fluorescent PSC-Derived Cardiomyocyte Reporter Lines: Generation Approaches and Their Applications in Cardiovascular Medicine"

_biology, 2020, doi:10.3390/biology9110402_

Round 1

Reviewer 1 Report

Sontayananon et al. provide a detailed review of genetic labeling of synthetic cardiomyocytes primarily for identification of distinct terminal lineages and then explore developing tools including optogenetics and calcium/voltage sensors for further specification. The manuscript is concise and employs a strategic approach to surveying the literature for fluorescent reporters after an abbreviated introduction to transgenics. The selected articles for review seems sufficient in most cases. The discussion of pre-/clinical utility for cardiac tissue repair provides a direction forward for the field. Overall, the manuscript has potential for broad impact to the Biology readership with interests in pluripotency and cardiomyocytes; however, a few significant concerns should be addressed:

[References] A single reference for published fluorescent PSC-CM reporter systems seems lacking given this is a review of the field. More citations will provide some sense of the ‘popularity’ or ‘how recent’ different models are relative to each other.

[Stem cell review] Induced pluripotency is suggested to be equivalent however the preparation of PSCs can lead to variable result. In addition, PSCs are not as a rule indefinite and can have alterations with passage number. While not the focus of the article some acknowledgement of the challenges of PSCs is warranted given that it is the cell source for the article.

[Abbreviations] A key is needed for all the abbreviation. The article seems targeted in part to an audience new to molecular biology and barring picking up Alberts textbook (which seems like it would have been a reference) both initialized abbreviations (e.g. FP) and non-initialized (e.g. ORF) should be consolidated in a key.

[Seemingly conflicts] Myosin heavy chain beta and Myosin light chain 2 (MYL2) are both listed as maturation markers. How do the two compare? Fig3, in text the Tbx5 and Nkx are listed as if binary, yet the figure suggests gradation of signal. Please clarify.

Minor:

[For future applications] In the conclusion, “advance[s] in 3D PSC-CM culture” is highlighted as an important topic though not discussed in any capacity in the article. The authors also mention disease without discussing in the article. Please address.

Author Response

Point 1 [References] A single reference for published fluorescent PSC-CM reporter systems seems lacking given this is a review of the field. More citations will provide some sense of the ‘popularity’ or ‘how recent’ different models are relative to each other.

Response 1 We are unsure how to respond to this comment, since the review already contains a comprehensive overview of the previous published reports of fluorescent PSC-CM reports systems – we would respectively point the reviewer to Table 2, which presents a summary of the reviewed literature in the manuscript. If we have missed crucial references, could the reviewer please point us to them, and we will add these to the manuscript.

Point 2 [Stem cell review] Induced pluripotency is suggested to be equivalent however the preparation of PSCs can lead to variable result. In addition, PSCs are not as a rule indefinite and can have alterations with passage number. While not the focus of the article some acknowledgement of the challenges of PSCs is warranted given that it is the cell source for the article.

Response 2 We thank the reviewer for raising this point and have extended the paragraph introducing PSCs (lines 50-57) to include information about artefacts of reprogramming and the impact of passage number on differentiation ability.

Point 3 [Abbreviations] A key is needed for all the abbreviation. The article seems targeted in part to an audience new to molecular biology and barring picking up Alberts textbook (which seems like it would have been a reference) both initialized abbreviations (e.g. FP) and non-initialized (e.g. ORF) should be consolidated in a key.

Response 3 We apologize for the heavy use of abbreviations – we have re-examined our text and removed examples of abbreviations where the term was only used a couple of times. That increases the clarity of the text. In addition, we present a list of abbreviations used, as suggested by the reviewer, which can be included as a box or a table, depending upon the journal formatting.

Point 4 [Seemingly conflicts] Myosin heavy chain beta and Myosin light chain 2 (MYL2) are both listed as maturation markers. How do the two compare?

Response 4 Both these genes (MYH7 and MYL2) are indeed both considered maturation factors – RNAseq datasets have revealed both of these genes to become significantly upregulated in iPSC-CMs from 1 week of age to 4 weeks of age, with MYL2 showing a more pronounced upregulation (Piccini et al., 2015). We have added a statement on this together with this reference to make this detail clearer in the section on functional ventricular iPSC-CMs (Lines 313-315). In addition, we explain in more detail the switching of MYH6 and MYH7 expression as iPSC-CMs mature and the species specific difference that are found (lines 286-289).

Point 5 Fig3, in text the Tbx5 and Nkx are listed as if binary, yet the figure suggests gradation of signal. Please clarify.

Response 5 Figure 3 was shown with a graduation of signal as the diagram was representing a FACs plot where an arbitrary threshold of expression is set by gating. Cells expressing below this arbitrary threshold are considered negative and cells expressing above this arbitrary threshold are considered positive for the particular marker. It is clear from the reviewer’s comment that this was unclear, so we have replaced the scale graduation with labels in the 4 quadrants which show a more binary outcome (as defined by the thresholds set in the FACs analysis).

Point 6 [For future applications] In the conclusion, “advance[s] in 3D PSC-CM culture” is highlighted as an important topic though not discussed in any capacity in the article. The authors also mention disease without discussing in the article. Please address.

Response 6 We thank the reviewer for this comment and now include a section on 3D iPSC-CM culture in a new section 7 on maturation approaches (lines 439-467).

Concerning disease, we have added a note on the importance of CM purification in disease modelling (lines 302-305 and 358). In the text, we highlight a number of examples where fluorescent reporters have benefited disease modelling – see Table 3 and examples in text (lines 378-379, 396-398, 406-408)

Reviewer 2 Report

In the proposed manuscript Sontayananon and colleagues reviewed the various methods that can be used to generate pluripotent stem cell-derived cardiomyocyte reporter lines. The authors extensively discussed the importance of this approach together with pro and cons of different approaches and highlighted their applications in cardiovascular medicine. The review is well structured, with tables and figures that facilitate the reader through the manuscript. I suggest minor revisions:

  • Please, extend the discussion with pertinent references about the non-cardiomyocyte cell types obtain with the current methods of pluripotent stem cell (PSC) cardiac differentiation, as this topic is mentioned several times in the manuscript (in the introduction and in paragraph 3).
  • The authors reviewed several methods of PSC-cardiomyocyte purification. Among the others, they mentioned reporter lines carrying an antibiotic resistant cassette under the control of a cardiac-specific gene’s regulatory elements e.g. MYH6. Please, update this statement with more recent literature. Also, please extend discussion about advantages and disadvantages of this approach supported by appropriated references.
  • Recent advances in techniques to obtain more mature PSC-cardiomyocyte phenotype and in turn advanced cardiac functionality have been made by engineering PSC-cardiac tissues. I recommend the authors to take into account and include in the manuscript these new 3D techniques that are substantially contributing to the cardiovascular field. For instance, PSC-cardiomyocyte-derived engineered heart tissues transduced with calcium sensors have been recently reported, also optogenetic approaches in PSC-cardiomyocyte-derived engineered cardiac tissues have recently been proposed.

Author Response

Point 1 Please, extend the discussion with pertinent references about the non-cardiomyocyte cell types obtain with the current methods of pluripotent stem cell (PSC) cardiac differentiation, as this topic is mentioned several times in the manuscript (in the introduction and in paragraph 3).

Response 1 We now specify the non-CM cell types (Line 254) and also have added a section to the text at lines 460-462 which addresses the effects of these cell types on differentiation and maturation in the context of 3D culture systems and we cite references describing the impact of each of the additional cell types. 

Point 2 The authors reviewed several methods of PSC-cardiomyocyte purification. Among the others, they mentioned reporter lines carrying an antibiotic resistant cassette under the control of a cardiac-specific gene’s regulatory elements e.g. MYH6. Please, update this statement with more recent literature. Also, please extend discussion about advantages and disadvantages of this approach supported by appropriated references.

Response 2 We have amended this section with a number of recent publications which explore the use of antibiotic resistance under the control of cardiac-specific gene’s regulatory elements – we address MYH6, TNNT2, NKX2.5 and NCX1 (Lines 268-269) and have updated Table 2. We also include the disadvantage of prolonged antibiotic treatments on cardiomyocyte electrophysiology (lines 269-272) and cite appropriate literature.

Point 3 Recent advances in techniques to obtain more mature PSC-cardiomyocyte phenotype and in turn advanced cardiac functionality have been made by engineering PSC-cardiac tissues. I recommend the authors to take into account and include in the manuscript these new 3D techniques that are substantially contributing to the cardiovascular field. For instance, PSC-cardiomyocyte-derived engineered heart tissues transduced with calcium sensors have been recently reported, also optogenetic approaches in PSC-cardiomyocyte-derived engineered cardiac tissues have recently been proposed.

Response 3 We thank the reviewer for this comment and now include a section on 3D iPSC-CM culture in a new section 7 on maturation approaches (lines 439-467), which addresses the points that include the combination of 3D culture with optogenetic and calcium sensor tools (Lines 459-467)

Reviewer 3 Report

Summary:

This work neatly described the possible methods to generate fluorescence reporter lines in PSCs. It further describes how such reporter lines can be applied for various studies at the example of cardiomyocyte biology.

Strength:

- very clearly written and organized, easy to read and understand

- beautiful, comprehensive figures

- good dose of background, yet focused on the topic

Major weaknesses:

none

Minor Weaknesses:

- table 1: it would help to write out the abbreviations in the table legend. gRNA for CRISPR is not mentioned, but the most widely used approach.

- the figures are beautiful, but the writing is very small

-fig. 1 legend, spelling out SHS and RMCE in legend would be helpful

- There are a lot of abbreviations throughout the text. For a non-cardiomyocyte reader, that makes it difficult to follow. To expand the usefulness of this review to other experts (which can easily apply), it might be nice to include an abbreviation summaery under key terms or a table or just reduce the use of them.

- a few sentences may be added to 6. about what the approaches are that have been made so far to further mature PSC-derived CMs and how these approaches may be enhanced with reporter lines.

Author Response

Point 1 table 1: it would help to write out the abbreviations in the table legend. gRNA for CRISPR is not mentioned, but the most widely used approach.

Response 1 We have reduced the number of abbreviations throughout the manuscript, and in particular, have reduced their use in figure legends. In addition, we now present a list of abbreviations that will help the clarity of our review. Concerning gRNA, we have chosen not to explain the CRISPR methodology as that is extensively covered by numerous other reviews, and hence avoid the use of gRNA and other CRISPR-related technical terms, again for reasons of clarity.

Point 2 - the figures are beautiful, but the writing is very small

Response 2 We have increased the font size, where possible, to aid in the clarity of the figures.

Point 3 fig. 1 legend, spelling out SHS and RMCE in legend would be helpful

Response 3  As mentioned above, we have removed all abbreviations from the figure legends

Point 4 There are a lot of abbreviations throughout the text. For a non-cardiomyocyte reader, that makes it difficult to follow. To expand the usefulness of this review to other experts (which can easily apply), it might be nice to include an abbreviation summaery under key terms or a table or just reduce the use of them.

Response 4 As mentioned above, we now include a list of abbreviations that help summarize the technical terms used.

Point 5 a few sentences may be added to 6. about what the approaches are that have been made so far to further mature PSC-derived CMs and how these approaches may be enhanced with reporter lines.

Response 5 We have added an additional section of maturation (Lines 439-467).